# Permeability of the blood-brain barrier through the phases of ischaemic stroke and relation with clinical outcome: protocol for a systematic review

Sara Bernardo-Castro ![ORCID],[1] Helena Donato,[2] Lino Ferreira,[3,4] João Sargento-Freitas ![ORCID] [1,4]

¹Stroke Unit, Centro Hospitalar e Universitário de Coimbra EPE, Coimbra, Portugal
²Documentation Service, Centro Hospitalar e Universitario de Coimbra EPE, Coimbra, Portugal
³Center for Neurosciences and Cell Biology, Universidade de Coimbra, Coimbra, Portugal
⁴Faculdade de Medicina, Universidade de Coimbra, Coimbra, Portugal

**Correspondence to**
Dr João Sargento-Freitas;
jsargentof@hotmail.com

## ABSTRACT

**Introduction** Ischaemic stroke is the most prevalent type of stroke and is characterised by a myriad of pathological events triggered by a vascular arterial occlusion. Disruption of the blood-brain barrier (BBB) is a key pathological event that may lead to fatal outcomes. However, it seems to follow a multiphasic pattern that has been associated with distinct biological substrates and possibly contrasting outcomes. Addressing the BBB permeability (BBBP) along the different phases of stroke through imaging techniques could lead to a better understanding of the disease, improved patient selection for specific treatments and development of new therapeutic modalities and delivery methods. This systematic review will aim to comprehensively summarise the existing evidence regarding the evolution of the BBBP values during the different phases of an acute ischaemic stroke and correlate this event with the clinical outcome of the patient.

**Methods and analysis** We will conduct a computerised search on Medline, EMBASE, Cochrane Central Register of Controlled Trials, Scopus and Web of Science. In addition, grey literature and ClinicalTrials.gov will be scanned. We will include randomised controlled trials, cohort, cross-sectional and case-controlled studies on humans that quantitatively assess the BBBP in stroke. Retrieved studies will be independently reviewed by two authors and any discrepancies will be resolved by consensus or with a third reviewer. Reviewers will extract the data and assess the risk of bias of the selected studies. If possible, data will be combined in a quantitative meta-analysis following the guidelines provided by Cochrane Handbook for Systematic Reviews of Interventions. We will assess cumulative evidence using the Grading of Recommendations, Assessment, Development and Evaluation approach.

**Ethics and dissemination** Ethical approval is not needed. All data used for this work are publicly available. The result obtained from this work will be published in a peer-reviewed journal and disseminated in relevant conferences.

**PROSPERO registration number** CRD42019147314.

## Strengths and limitations of this study

► To our knowledge this is the first systematic review that will focus on the progression of the blood-brain barrier permeability (BBBP) during acute ischaemic stroke (AIS) and its clinical consequences.
► This protocol has been developed following Preferred Reporting Items for Systematic Review and Meta-Analysis Protocols.
► This systematic review could help to guide conventional recanalisation treatments outside the therapeutic windows and other innovative delivery treatments in later stage of AIS.
► We will include all types of studies, not imposing any restriction on language or year.
► The vast heterogeneity that can arise from the use of different BBBP imaging techniques, the pharmacokinetic models used and the different permeability parameters yield from each technique, may prevent a quantitative meta-analysis from being conducted.

year around 14 million people suffer a stroke, 5.5 million of which die[1] and another 5 million stay permanently disabled, representing a significant concern for public health and society.[2] Acute ischaemic stroke (AIS) accounts for approximately 85% of all strokes,[1 3 4] it restricts blood flow to a specific region of the brain leading to death of the compromised tissue.[5 6] Currently, the only available and effective treatment to limit this situation is recanalisation therapy, to restore the normal blood flow,[7] but these therapies can only be given to less than 5% of patients due to their narrow therapeutic window.[4 7] Treating patients outside this window could contribute to additional tissue damage and increase in the risk of haemorrhagic transformation (HT).[7 8]

The blood-brain barrier (BBB) is a dynamic physiological structure that constitutes an interface between the vasculature system and the neural tissues. It regulates the transport of

## INTRODUCTION

Stroke is the second leading cause of mortality and morbidity worldwide.[1–4] Every

substances in a bidirectional way[9] and protects the central nervous system from unwanted compounds, playing a crucial role in maintaining its homeostasis.[9 10] During the process of ischaemia the BBB undergoes a dysfunction[9 11] that leads to an increase of its permeability,[10] enabling the passage of large molecules, fluids and blood into the brain interstitium.[8 9] This pathological leakage is associated with a worst outcome after AIS[3 8 12] and is known to persist for several days[7] following a time-course mediated by complex pathophysiological events with different clinical implications.[13–15] During the first hours after stroke, namely the hyperacute and acute phases, the insult triggers ischaemic cell death which leads to a higher risk of HT.[16 17] In a late acute/early subacute stage, the BBB is disrupted due to the secondary ischaemic injury which causes inflammatory cell infiltration and tissue scarring and a further BBB permeability (BBBP) increase.[17 18] In a later subacute stage, the BBBP relies on physiological recovery processes such as neoangiogenesis, as demonstrated in both animals[19 20] and humans.[21] Whereas the existence of this permeability process is unequivocal, its concrete evolution is not yet certain.

Several longitudinal animal studies have tried to explain this event. Some studies point to an 'open-close-open' biphasic pattern in which the BBB has increased permeability at a first stage followed by a return to normal values and a second permeability increase.[22–24] Nonetheless these studies show differences on the open/close times and more recent literature points to a more continuous opening of the BBB with biphasic permeability peaks but without total closing.[25–28] A first BBBP increase has been shown to appear as soon as 3–6 hours after occlusion, followed by a decrease but not a total recovery, and a second peak at the early subacute stage.[26 27 29] Studies extending the BBBP quantification time-points have reported a further increased permeability up to 1,[27 29 27]3 and even 5 weeks[28 30] after occlusion, suggesting that the BBB could remain open until months after the onset of stroke.

Very few human studies have focused on studying this evolution[11 21 31 32] and even though these studies point to a continuous opening of the BBB, they do not offer a clear and collective evidence on the magnitude of this opening in the different phases. A quantitative assessment of the BBB disruption through its permeability could add valuable information in the evaluation of patients with AIS.

Three main imaging techniques are used to evaluate BBBP: CT, MRI and positron emission tomography (PET).[33] Nonetheless, due to the limited availability of PET, the 'permeability imaging' term is used mainly for MRI and CT.[7] These specific imaging tools are able to measure the BBBP in vivo in a non-invasive manner.[3 9 11 34–37] In short, these imaging modalities quantify the rate and amount at which a specific contrast agent leaves the blood stream and enters the brain parenchyma[12 34 38] using mathematical models able to describe the physiological characteristics of the BBB such as vessel permeability, vessel surface area product and tissue

volume fraction.[11 39 40] In clinical practice this information has been used as a diagnostic tool for differential diagnosis, and also to support decisions for safer and improved recanalisation therapies for patients who had a stroke in an extended time window.[7–9 11 38 41–43]

Nonetheless, although there are important systematic reviews that focus on the implications of imaging and increased permeability on stroke outcome,[44 45] and on the utility of perfusion imaging in determining treatment eligibility in patients who had an acute stroke,[46] to date there are no systematic reviews, to our knowledge, focusing on the development of the BBBP during the phases of AIS.

Ideally, this knowledge would help not only in extending the treatment window, but also in the development of future treatment options such as delivery system strategies for neuroprotective or neurorestorative treatments that aim to use the BBB as a therapeutic vehicle or target. Therefore, there is a need to perform a systematic review and meta-analysis on the BBBP dynamics after AIS to gather larger sample sizes of patients and create a concrete understanding of the subject. This systematic review will provide an insight on the evolution of the permeability of the BBB in patients affected by AIS through the different stages of the stroke and its relevance in the patient outcome and treatment.

## Objective

The main objective of this work is to carry out a systematic review and meta-analysis on the BBBP during the different phases of an acute ischaemic stroke with the aim of assessing its evolution through time and its correlation with clinical outcome.

## METHODS

### Eligibility criteria

This work will identify randomised controlled trials (RCTs), cohort studies (prospective or retrospective), cross-sectional studies and case-controlled studies that quantify BBBP in patients suffering from AIS. Studies fulfilling the eligibility criteria shown in table 1 will be selected for further review. If more than one article reports the same study, the article with the largest sample size or reporting more relevant data for our specific aim will be selected. No restriction regarding publication year will be set; therefore, we will be including studies since inception to 31st of July 2019. In addition, no language restriction will be applied. If a study in a non-understandable language is obtained, we will consider its suitability for our study by its English abstract and if the information is interesting enough to be included, the paper will be sent to a professional translator.

### Information sources

We will conduct a comprehensive computerised literature search strategy to find the studies that will take part in this systematic review. We will search for both published

**Table 1** Inclusion/exclusion criteria for study selection

| Inclusion criteria | Exclusion criteria |
|---|---|
| Studies on living humans | Non-human studies |
| Acute ischaemic stroke | Lacunar strokes (subcortical ischaemic lesion with a diameter under 15 mm in CT or 20 mm in MRI) |
| BBBP evaluation through neuroimaging | Mild stroke (NIHSS below 6) |
| Studies with a follow-up for clinical outcome | Haemorrhagic stroke |
| | BBBP evaluation through non-imaging techniques |
| | BBBP evaluation in other non-AIS diseases |
| | No primary research |
| | Reports just defining a study protocol |
| | Case-report studies |
| | Studies not reporting time from onset to imaging |
| | Studies not reporting contralateral permeability values |

AIS, acute ischaemic stroke; BBBP, blood-brain barrier permeability; NIHSS, National Institutes of Health Stroke Scale.

and unpublished studies in the following databases: PubMed/Medline, EMBASE, Web of Science, Scopus and Cochrane Central Register of Controlled Trials. Other electronic platforms such as ClinicalTrials.gov will be scanned to keep up with ongoing or unpublished clinical trials. If any relevant unpublished trial is found, the corresponding author listed will be contacted to obtain the required information. If no response is given or, if the author decides not to share the data, this will be listed as the reason for exclusion of said trial. In addition to this electronic search, a supplementary search of the grey literature will be conducted with the aim of including all possible existing articles on the subject. No pre-prints will be included on the study.

## Search strategy
The search strategy will include the following terms and all of its variants in multiple combinations adapted to each one of the databases regarding its own special requirements as shown in table 2: 'stroke', 'permeability', 'blood brain barrier', 'imaging', 'neuroimaging'. The search of the grey literature will include a by-hand search of relevant articles in the listed bibliography of the selected studies and important reviews on the subject, conference papers and a Google search of the used terms.

## Data management
All publications arising from the literature search conducted will be imported to the Mendeley citation software where duplicates will be managed and erased and titles/abstracts of all records will be scanned.

## Selection process
Two independent reviewers will conduct the selection process. All records identified in the search stage will be screened by title/abstract and studies clearly not matching the criteria will be discarded. The remaining studies will be full-text reviewed and included or discarded according to the inclusion/exclusion criteria. Any disagreement between the reviewers will be resolved by consensus or by a third one if necessary. Reasons for the exclusion of full-text records will be recorded. Details on the selection process of the studies will be documented into a flow chart following the Preferred Reporting Items for Systematic Reviews and Meta-Analyses[47] as presented in figure 1.

## Data collection process
To ensure that all relevant information is captured, and to minimise the risk of bias, two reviewers will independently extract the information from the studies following the same pilot form. Any disagreement will be resolved by consensus. The data extracted will be reviewed and validated by a third reviewer.

## Data items
Four main categories of data will be extracted from all studies selected: (1) features of the study; (2) patients' characteristics; (3) intervention; (4) outcome. Among these categories a number of items will be collected as presented in table 3.

Since the main aim of this work is to study the BBBP values in the different phases of stroke, we will form the following groups according to time from onset to imaging reported in each study:
1. Hyperacute stage: 6 hours or less.
2. Acute stage: between 6 and 48 hours.
3. Subacute stage: between 3 and 9 days.
4. Chronic stage: 30 or more days.

For any study reporting more than one BBBP measurement, each of the measurements will be considered as an independent study and will be placed in the corresponding group according to the time-points established above. These values will be identified as author, year followed by the name of the corresponding stage.

## Outcomes and prioritisation
This work has three primary outcomes: (1) the comparison of the quantitative permeability values across time after stroke; (2) the association between the different BBBP values and the functional outcome of patients who had an acute stroke; (3) association between permeability values and the recanalisation treatment given.

When and if possible, the following secondary outcomes will be measured: (1) the association between the different BBBP values and haemorrhagic transformation; (2)

**Table 2** Retrieval search strategy

**PubMed**

| Query | Search |
|---|---|
| #1 | 'Stroke' (MeSH terms) OR 'stroke' OR 'cerebral stroke' OR 'ischemic stroke' OR 'acute stroke' OR 'acute ischemic stroke' OR 'apoplexy' OR 'cerebral apoplexy' OR 'cerebrovascular accident' OR 'acute cerebrovascular accident' OR 'brain vascular accident' OR 'CVA' |
| #2 | 'Blood-Brain Barrier' (MeSH terms) OR 'Blood-Brain Barrier' OR 'Blood Brain Barrier' OR 'Brain-Blood Barrier' OR 'Hemato Encephalic Barrier' OR 'Hemato-Encephalic Barrier' |
| #3 | 'Permeability' (MeSH terms) OR 'Permeability' OR 'Leakage' |
| #4 | 'Diagnostic Imaging' (MesH) OR 'Neuroimaging' (MeSH Terms) OR 'Neuroimaging' OR 'Brain Imaging' OR 'Magnetic Resonance Imaging' (MeSH Terms) OR 'Magnetic Resonance Imaging' OR 'MRI' OR 'MRI scan' OR 'Functional MRI' OR 'fMRI' OR 'Computed Tomography Angiography' (MeSH Terms) OR 'Computed Tomography Angiography' OR 'Computed Tomography' OR 'CT' OR 'CT angiography' OR ' dynamic contrast enhanced MRI' OR 'dynamic susceptibility contrast MRI' OR 'computed tomography perfusion' |
| #5 | Search #1 AND #2 AND #3 AND #4 |

**EMBASE**

| | |
|---|---|
| #1 | 'cerebrovascular accident'/exp OR 'brain ischemia'/exp OR 'cerebrovascular accident' OR 'stroke patient' OR 'brain ischemia' OR 'stroke' OR 'acute ischemic stroke' OR 'ischemic stroke' OR 'apoplexy' OR 'cerebral apoplexy' OR 'brain apoplexy' |
| #2 | 'blood brain barrier'/exp OR 'blood brain barrier' OR 'hemato encephalic barrier' OR 'hemato-encephalic barrier' |
| #3 | 'permeability'/exp OR 'permeability' OR 'leakage' |
| #4 | 'diagnostic imaging'/exp OR 'neuroimaging'/exp OR 'functional magnetic resonance imaging/exp OR 'nuclear magnetic resonance imaging/exp OR 'nuclear magnetic resonance'/exp OR 'computer assisted tomography'/exp OR 'neuroimaging' OR 'nuclear magnetic resonance imaging OR 'mri' OR 'functional magnetic imaging' OR 'fmri' OR 'computer assisted tomography' OR 'computed tomographic angiography' OR 'ct' OR 'dynamic contrast enhanced MRI' OR 'dynamic susceptibility contrast MRI' OR 'computed tomography perfusion' |
| #5 | #1 AND #2 AND #3 AND #4 |

**Scopus**

| | |
|---|---|
| #1 | TITLE-ABS-KEY ('stroke') OR TITLE-ABS-KEY ('ischemic stroke') OR TITLE-ABS-KEY ('acute ischemic stroke') OR TITLE-ABS-KEY ('cerebral apoplexy') OR TITLE-ABS-KEY ('cerebrovascular accident') OR TITLE-ABS-KEY ('acute cerebrovascular accident') OR TITLE-ABS-KEY ('brain apoplexy') OR TITLE-ABS-KEY ('CVA') |
| #2 | ('blood-brain barrier') OR TITLE-ABS-KEY ('hemato encephalic barrier') OR TITLE-ABS-KEY ('blood brain barrier') OR TITLE-ABS-KEY ('hemato-encephalic barrier') |
| #3 | TITLE-ABS-KEY ('permeability') OR TITLE-ABS-KEY ('leakage') |
| #4 | TITLE-ABS-KEY ('neuroimaging') OR TITLE-ABS-KEY ('magnetic resonance imaging') OR TITLE-ABS-KEY ('functional magnetic resonance imaging') OR TITLE-ABS-KEY ('MRI') OR TITLE-ABS-KEY ('fMRI') OR TITLE-ABS-KEY ('computed tomography') OR TITLE-ABS-KEY ('computed tomography angiography') OR TITLE-ABS-KEY ('CT') OR TITLE-ABS-KEY ('dynamic contrast enhanced MRI') OR TITLE-ABS-KEY ('dynamic susceptibility contrast MRI') OR TITLE-ABS-KEY ('computed tomography perfusion') |

**Cochrane Central Register of Controlled Trials**

| | |
|---|---|
| #1 | ('blood brain barrier' OR 'hemato encephalic barrier' OR 'hemato-encephalic barrier') AND ('permeability' OR 'leakage') AND ('cerebrovascular accident' OR 'stroke patient' OR 'brain ischemia' OR 'stroke' OR 'acute ischemic stroke' OR 'ischemic stroke' OR 'apoplexy' OR 'cerebral apoplexy' OR 'brain apoplexy') AND ('neuroimaging' OR 'nuclear magnetic resonance imaging OR 'mri' OR 'functional magnetic imaging' OR 'fmri' OR 'computer assisted tomography' OR 'computed tomographic angiography' OR 'ct' OR 'dynamic contrast enhanced MRI' OR 'dynamic susceptibility contrast MRI' OR 'computed tomography perfusion') |

**Web of Science**

| | |
|---|---|
| #1 | TS=('blood brain barrier' OR 'hemato encephalic barrier' OR 'hemato-encephalic barrier') AND TS=('permeability' OR 'leakage') AND TS=('cerebrovascular accident' OR 'stroke patient' OR 'brain ischemia' OR 'stroke' OR 'acute ischemic stroke' OR 'ischemic stroke' OR 'apoplexy' OR 'cerebral apoplexy' OR 'brain apoplexy') AND TS=('neuroimaging' OR 'nuclear magnetic resonance imaging OR 'mri' OR 'functional magnetic imaging' OR 'fmri' OR 'computer assisted tomography' OR 'computed tomographic angiography' OR 'ct' OR 'dynamic contrast enhanced MRI' OR 'dynamic susceptibility contrast MRI' OR 'computed tomography perfusion') |

MeSH, Medical Subject Headings.

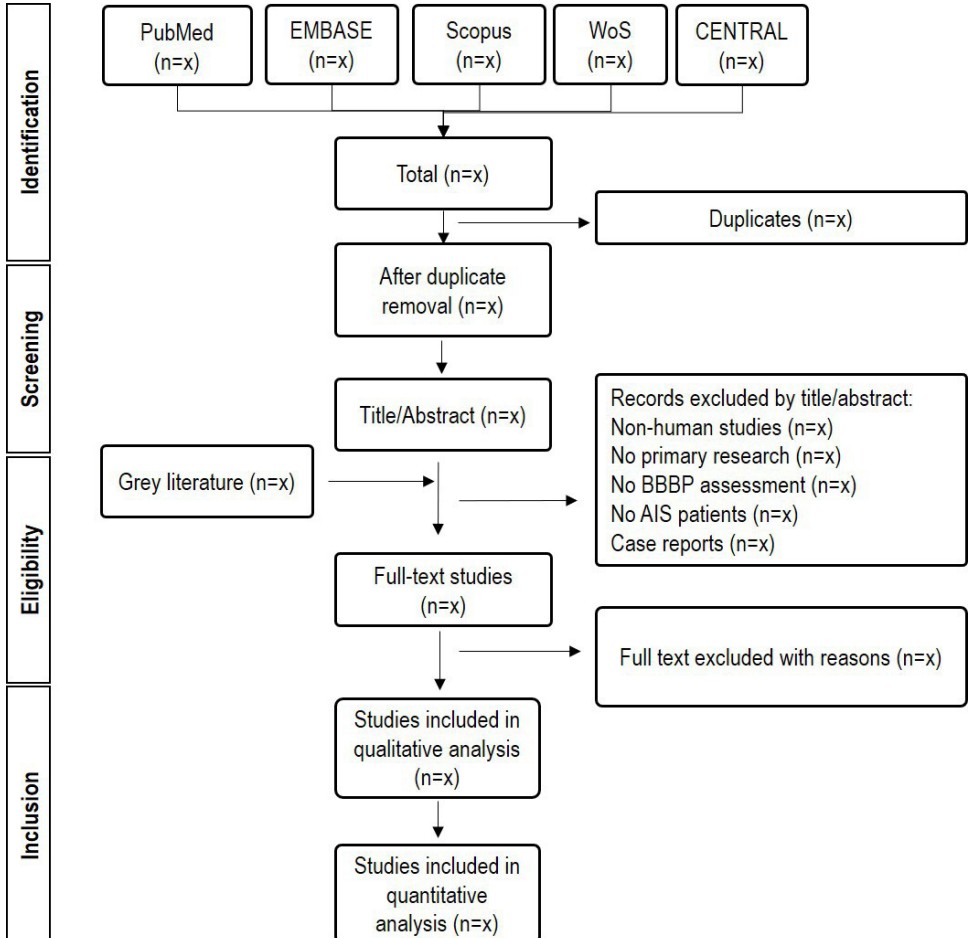

**Figure 1** Flow chart diagram presenting the selection process for the studies. AIS, acute ischaemic stroke; BBBP, blood-brain barrier permeability; CENTRAL, Cochrane Central Register of Controlled Trials.

association between any clinical feature/stroke predictor (age, hypertension, diabetes) and the BBBP.

### Risk of bias in individual studies

With the aim of minimising bias, the methodological quality of all studies included in the systematic review, will be assessed independently by two reviewers. Since we will be including diverse types of studies, we will use different tools to assess the risk of bias depending on the characteristics of the studies, tuning these tools if necessary. For the RCT we will be using the Cochrane Collaboration's tool for assessing risk of bias in randomised trials.[48] This tool covers seven sources of bias: (1) random sequence generation; (2) allocation concealment; (3) blinding of participants and personnel; (4) blinding of outcome assessment; (5) incomplete outcome data; (6) selective reporting and (7) other bias. The risk of bias for each domain will be graded as high, low or unclear based on the relevant information extracted from each study. Low risk of bias will be given to the study if all of the domains are marked as low risk; intermediate risk of bias will be

| Table 3 | Data items to be collected from the selected studies | | |
|---|---|---|---|
| **Features of the study** | **Patients' characteristics** | **Intervention** | **Outcome** |
| Title, author | Age, gender | Time from onset to imaging | Permeability values |
| Study design | Comorbidities | Imaging characteristics | Final lesion (volume) |
| Recruitment procedure and duration | NIHSS at admission | BBBP assessment characteristics | Follow-up (length, number) |
| Number of participants | Stroke aetiology (TOAST classification) | Reperfusion treatment given | Clinical outcome (NIHSS and mRS) |
| Imaging modality | Vascular territory | | Haemorrhagic transformation |

BBBP, blood-brain barrier permeability; mRS, Modified Rankin Scale; NIHSS, National Institutes of Health Stroke Scale; TOAST, Trial of ORG 10172 in Acute Stroke Treatment.

given when at least one of the domains is graded with unclear risk; high risk of bias will be given if high risk is given to at least one of the domains of the checklist.

For non-randomised trials we will use the Newcastle-Ottawa Scale for assessing the quality of non-randomised studies in meta-analysis.[49] These studies will be assessed based on three perspectives: (1) selection of study groups, (2) comparability of the groups; (3) ascertainment of exposure (in case–control studies) or outcome of interest (in cohort studies). This scale proposes a system in which a high-quality choice will be granted by a star. A maximum of 9 stars for study can be given. We will consider a score of 7 or more as low risk of bias/high-quality and less than 5 will be considered as high risk of bias/poor quality.[50 51]

Any disagreement between the two reviewers will be resolved by consensus or by a third reviewer if necessary.

### Data synthesis

This systematic review will include a quantitative meta-analysis if possible. The statistical analysis will be carried out taking into account the guidelines provided by Cochrane Handbook for Systematic Reviews of Interventions.[48] As our main outcome will be presented as continuous data (permeability values), we will use the mean difference or the standardised mean difference and the respective 95% CI to combine the results. We will test the consistency and heterogeneity of the studies with the Higgins $I^2$ statistic that can also be used to describe heterogeneity among subgroups.[52] Following the direction given by Higgins *et al*[52] we will consider ≤25% as low heterogeneity, between 25% and 50% as moderate heterogeneity and >75% as high heterogeneity. If the $I^2$ value is ≤50% (low to moderate heterogeneity), we will use the fixed effect model for data synthesis; if it is greater than 50%, we will use the random effects model. If the heterogeneity values are over 75%, we will search for the possible sources of this high heterogeneity, including reviewing the methodological processes of the selected studies, and search for outliers or influential cases that may distort the results of the analysis. Any possible outlier or influential case, as well as studies presenting with poor methodological quality and/or a high or critical risk of bias, will be excluded in a further sensitivity analysis.

If we are not able to collect the appropriate outcome information or not enough studies are retrieved for the different stages, we will consider that a quantitative meta-analysis is not feasible and we will conduct a narrative description.[53]

### Subgroup analysis

When possible, the following subgroups will be made:
- ► Subgroups according to the imaging technique used with the aim of reducing possible heterogeneity arising from this methodological variety.
- ► Subgroups according to the treatment received.
- ► Subgroups according to the presence/absence of HT.
- ► Subgroups according to the mRS 90 days value: (1) mRS 0–1; (2) mRS 2–5

We will compare the permeability values among the subgroups and, if possible, correlate these with the different features/predictors of stroke.

### Meta-bias(es)

To assess publication bias, we will conduct a funnel plot following the recommendation of the Cochrane Handbook for Systematic Reviews of Interventions[48] and a complemental Egger's test in order to quantify the funnel plot's asymmetry.

### Confidence in cumulative evidence

The strength of the body evidence will be assessed using the Grading Recommendations Assessment, Development and Evaluation.[48]

### Patient and public involvement

This is a protocol for a systematic review that will be based on previously published data, therefore no participant recruitment will take place. The involvement of participants on the recruitment and dissemination of results is not applicable.

## ETHICS AND DISSEMINATION

This work will be based in data that are public and already published, therefore an ethical approval would not be necessary. The result obtained from this work will be published in a peer-reviewed journal and disseminated in relevant conferences. If any amendments are needed due to deviations from this protocol in the execution of the study, these amendments will be recorded and noted in the publication.

**Contributors** JSF and SBC formulated the idea for this systematic review. SBC drafted the protocol guided by JSF and HD. JSF, HD and LF reviewed all manuscript versions.

**Funding** This work is framed within the NANOSTEM project. This project has received funding from the European Union's Horizon 2020 research and innovation program under grant agreement number 764958.

**Competing interests** None declared.

**Patient and public involvement** Patients and/or the public were not involved in the design, or conduct, or reporting, or dissemination plans of this research.

**Patient consent for publication** Not required.

**Provenance and peer review** Not commissioned; externally peer reviewed.

**ORCID iDs**
Sara Bernardo-Castro http://orcid.org/0000-0001-8809-5101
João Sargento-Freitas http://orcid.org/0000-0003-4665-5697

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
