## [Reviewer comments · BMJ Open]

ARTICLE DETAILS

TITLE (PROVISIONAL)	Permeability of the blood brain barrier through the phases of ischemic stroke and relation with clinical outcome: protocol for a systematic review
AUTHORS	Bernardo-Castro, Sara; Donato, Helena; Ferreira, Lino; Sargento-Freitas, João

VERSION 1 – REVIEW

REVIEWER	Richard Leigh Johns Hopkins University USA
REVIEW RETURNED	01-May-2020

GENERAL COMMENTS	In their manuscript “Permeability of the blood brain barrier through the phases of ischemic stroke and relation with clinical outcome: protocol for a systematic review,” Bernardo-Castro et al. describe a protocol for performing a literature review of studies looking at the role of BBB disruption in human ischemic stroke. There is a brief description of how the BBB is affected in various time frames after stroke which is followed by the stated objective of carrying out “a systematic review and meta-analysis on the evolution of the permeability values of the blood-brain barrier during the different phases of an acute ischemic stroke and correlate this event with the clinical outcome of the patient.” They then lay out the methods for identifying the literature that will be used, the type of data to be extracted, how it will be synthesized and approaches to avoiding bias. While reviewing the literature on BBB and stroke is a very worthwhile endeavor, I have some trouble understanding purpose of this study protocol. On the one hand it should give more detail about what is known about the BBB prior to their literature review, on the other hand that type of information may be more appropriate for the manuscript that describes the study proposed in the protocol. There are several issues with the manuscript both in the language used and the content included, or lack thereof, as outlined below: 1. There are several issues that are either due to improper English or due to typos thus the paper should be edited by a native English speaker. Examples:a. Line 24-25 “reviewers will scan the studies” could be phrased a couple different ways but this is not one of themb. Line 25 “will be solved by consensus” should be “will be resolved by consensus”c. Line 54 “outcomes could difficult the performance”2. The introduction does not lay out what is known and what is unknown, particularly that which has been worked out in the animal literature.
--

	3. The introduction does not talk about what imaging methods exist for use in humans (CT,MRI,PET). 4. The idea of performing a meta-analysis has a lot of challenges that are not addressed in this protocol. For instance, most hyperacute BBB measurements are done using DSC MRI, whereas acute BBB measurements use both DCE MRI and DSC MRI. Are these two methods comparable? What about CT perfusion? How are these various methods going to be combined? If they disagree, which one is more reliable? 5. This review is actually covering two topics:  hyperacute pre-treatment BBB imaging to make reperfusion therapy decisions Subacute changes in BBB and clinical outcome If that is correct, then this should be stated. If it is incorrect then the paper should be clearer about what else the review is studying.
--	---

REVIEWER	Ahmed Khalil Charité Universitätsmedizin Berlin, Germany Berlin Institute of Health (BIH), Germany Humboldt Universität zu Berlin, Germany Max Planck Institute for Human Cognitive and Brain Sciences, Germany
REVIEW RETURNED	05-May-2020

GENERAL COMMENTS	In this protocol, the authors propose conducting a systematic review of the evolution of blood brain barrier (BBB) permeability over time assessed using neuroimaging in human stroke patients, as well as its relationship to clinical outcome. It is my opinion that the answers to these questions would be very relevant to enhancing our understanding of stroke pathophysiology, and would provide crucial information for the planning and interpretation of future interventional stroke therapy studies. The major challenge that I foresee, which in my view does not preclude that this study be done, is the considerable heterogeneity of the literature on this topic (see for example Suh et al., European Radiology 2019, Heye et al., Neuroimage Clinical 2014, Farrall & Wardlaw, Neurobiology of Aging 2009). Major comments  1. It is crucial for the proposed study that the authors define exactly what they mean by "quantitative" BBB permeability assessment . There is little consensus in the literature on what this means. Particularly, the authors should elaborate on which combination(s) of imaging modalities, pharmacokinetic models, and permeability parameters they consider as providing a "quantitative" assessment. 2. In the "Data Management" section, the authors describe how the found studies will be managed. How will the data extracted from eligible studies be handled? Where and how will it be stored? What kind of program(s) will be used to analyze the data? 3. It would be very beneficial if the authors could provide an example of the data collection form with the protocol (see Minor Comments 8-11 below). 4. The authors describe how the risk of bias / methodological quality of the eligible studies will be assessed. How will this assessment influence, or be incorporated into, the review? Will studies below a certain quality threshold be excluded from the meta-analysis? Will a sensitivity analysis be performed? Or will the findings of the studies be placed in the context of their methodological quality?
--

	5. What criteria will be used to determine whether or not a quantitative meta-analysis will be feasible (page 9, line 55)? Will this be based on statistical heterogeneity, clinical heterogeneity, both, or other factors (see Ioannidis et al., BMJ 2008)? Minor comments  1. The systematic review by Ryu et al. (cited on page 5, line 39) dealt with hypoperfusion and penumbral selection for reperfusion therapy in acute stroke, and did not investigate the influence of BBB permeability. 2. How do the authors define "lacunar" and "mild" strokes (page 6, line 28)? Will a size / threshold criterion be used to define a lacunar stroke? Will an NIHSS threshold be used to define mild strokes? 3. It's not clear why the authors are planning to exclude "Studies no [sic] reporting contralateral permeability values" if they are only including studies with quantitative BBB permeability values (although this depends on how they define "quantitative", see Major Comment 1). 4. How will studies in languages that the investigators do not understand be dealt with? 5. Will the authors include preprints in their search strategies? If so, which databases will be searched for preprints? 6. How will information from unpublished studies found on ClinicalTrials.gov be incorporated into the review? 7. The authors might want to consider expanding their search terms to include "leakage", as well as the individual neuroimaging modalities / techniques they are looking for (e.g. "dynamic contrast enhanced MRI", "dynamic susceptibility contrast MRI", "computed tomography perfusion"). 8. What is meant by "Type of stroke" in Table 3? Does this refer to the stroke etiology? What kind of classification system will be used to assess this (e.g. TOAST)? 9. Which "imaging characteristics" (Table 3) will be included? 10. By "treatment given" (Table 3), are the authors referring to stroke-specific treatments in the acute phase (i.e. intravenous thrombolysis and mechanical thrombectomy)? Or does this variable also include non-stroke-specific and later treatments (e.g. secondary prevention therapy)? 11. Which "permeability values" (Table 3) will be extracted (see also Major Comment 1). 12. Will the grouping based on time from onset to imaging (page 8, lines 53-57) be done by study or within each study as well (if this information is available)? 13. Will statistical tests be used in addition to the funnel plot to assess publication bias?
--	--

VERSION 1 – AUTHOR RESPONSE

REVIEWER 1

Reviewer Name: Richard Leigh

Institution and Country: Johns Hopkins University, USA

Please state any competing interests or state 'None declared': None declared

In their manuscript "Permeability of the blood brain barrier through the phases of ischemic stroke and relation with clinical outcome: protocol for a systematic review," Bernardo-Castro et

al. describe a protocol for performing a literature review of studies looking at the role of BBB disruption in human ischemic stroke. There is a brief description of how the BBB is affected in various time frames after stroke which is followed by the stated objective of carrying out “a systematic review and meta-analysis on the evolution of the permeability values of the blood-brain barrier during the different phases of an acute ischemic stroke and correlate this event with the clinical outcome of the patient.” They then lay out the methods for identifying the literature that will be used, the type of data to be extracted, how it will be synthesized and approaches to avoiding bias. While reviewing the literature on BBB and stroke is a very worthwhile endeavor, I have some trouble understanding purpose of this study protocol. On the one hand it should give more detail about what is known about the BBB prior to their literature review, on the other hand that type of information may be more appropriate for the manuscript that describes the study proposed in the protocol. There are several issues with the manuscript both in the language used and the content included, or lack thereof, as outlined below:

- 1. There are several issues that are either due to improper English or due to typos thus the paper should be edited by a native English speaker. Examples:**

Response: we thank the reviewer for pointing out these issues. Our manuscript has been revised to improve readability, and all changes have been highlighted in the manuscript. Below are the responses to each of the reviewer's suggestions.

- a. Line 24-25 “reviewers will scan the studies” could be phrased a couple different ways but this is not one of them**

Response: we corrected the sentence to: “Retrieved studies will be independently reviewed by two authors”. Page 2; lines 17-18.

- b. Line 25 “will be solved by consensus” should be “will be resolved by consensus”**

Response: corrected as suggested. Page 2-line 18; Page 7-line 6; Page 7-line 13; Page 8-line 23.

- c. Line 54 “outcomes could difficult the performance”**

Response: we corrected the sentence to: “[...] may prevent a quantitative meta-analysis from being conducted”. Page 2; lines 40-41

- 2. The introduction does not lay out what is known and what is unknown, particularly that which has been worked out in the animal literature.**

Response: we agree that the introduction should further include a deeper review on the current knowledge on the animal field in this topic. This information has been implemented in the manuscript. Page 3; lines 30-39. In addition, and in request of the reviewer's comment in the initial paragraph of the response “the protocol should give more detail about what is known about the BBB” we added a brief paragraph on the introduction (page 3; lines 12-15) about what is the BBB and its main function. Nonetheless and agreeing with the further comment of the reviewer, we think that a deeper view in this topic is more suitable for the SR-MA manuscript than for this protocol.

- 3. The introduction does not talk about what imaging methods exist for use in humans (CT,MRI,PET).**

Response: we agree with the reviewer that the introduction should further cover the human imaging methods for BBB assessment. We included this information in the new version of the manuscript. Page 4; lines 5-15.

- 4. The idea of performing a meta-analysis has a lot of challenges that are not addressed in this protocol. For instance, most hyperacute BBB measurements are done using DSC MRI,**

whereas acute BBB measurements use both DCE MRI and DSC MRI. Are these two methods comparable?

Response: we agree with the reviewer on the many challenges that are presented when performing a meta-analysis. In this case the reviewer points the possible inability to compare DCE and DSC MRI. These two methods are in fact difficult to pool and compare in a meta-analysis, but we do not aim to study the suitability of these methods as predictors of outcome or as clinical tools for treatment eligibility, instead, what we aim to do is a comparison on the increment of the permeability in the ipsilateral side when compared with the contralateral side on each study and then do an in-between study comparison matching by time-point of scan. This way although imaging modality will be of course the main source of heterogeneity we will not be comparing a single measure, but an increment on the permeability values. This is the reason that leads us to think that even if the imaging method is different in between studies, we can pool the permeability values, always addressing these problems in the review.

In light with the reviewer's comment we reinforced this limitation in page 2 of this protocol, "strengths and limitations of this study" lines 37-41.

5. This review is actually covering two topics:

- a. Hyperacute pre-treatment BBB imaging to make reperfusion therapy decisions**
- b. Subacute changes in BBB and clinical outcome**

If that is correct, then this should be stated. If it is incorrect then the paper should be clearer about what else the review is studying.

Response: We thank the reviewer for the comment. This SR-MA proposes to investigate the impact and associations of BBB permeability in the different phases of stroke, pooling together patients with BBB assessment performed at different stages, regardless of the indication. Some of the studies are in fact pre-reperfusion, others were performed in a post-reperfusion stage. As answered in comment 4, we do not aim to review BBB permeability measures for reperfusion therapy decisions neither prediction of outcome post-reperfusion. We aim to study the permeability dynamics through the different stroke stages of the disease regardless of any treatment. The reperfusion therapies will be, nonetheless, used as a subgroup analysis to explain part of the results, as pointed in the "subgroup analysis" section of the manuscript (page 9; lines 4-12), but the main aim of this work is to try to understand the dynamics that follow the permeability of the blood brain barrier in the development of stroke, since literature is quite contradictory on addressing this topic.

In light of the reviewer's comment we understand that our main objective was more ambiguous than intended, and therefore, we have adjusted the text to be clearer. Page 4; lines 30-33.

REVIEWER 2

Reviewer Name: Ahmed Khalil

Institution and Country: Charité Universitätsmedizin Berlin, Germany; Berlin Institute of Health (BIH), Germany Humboldt Universität zu Berlin, Germany, Max Planck Institute for Human Cognitive and Brain Sciences, Germany

Please state any competing interests or state 'None declared': None declared

In this protocol, the authors propose conducting a systematic review of the evolution of blood brain barrier (BBB) permeability over time assessed using neuroimaging in human stroke patients, as well as its relationship to clinical outcome. It is my opinion that the answers to these questions would be very relevant to enhancing our understanding of stroke pathophysiology, and would provide crucial information for the planning and interpretation of future interventional stroke therapy studies. The major challenge that I foresee, which in my view does not preclude that this study be done, is the considerable heterogeneity of the

literature on this topic (see for example Suh et al., European Radiology 2019, Heye et al., Neuroimage Clinical 2014, Farrall & Wardlaw, Neurobiology of Aging 2009). Major comments:

- 1. It is crucial for the proposed study that the authors define exactly what they mean by "quantitative" BBB permeability assessment. There is little consensus in the literature on what this means. Particularly, the authors should elaborate on which combination(s) of imaging modalities, pharmacokinetic models, and permeability parameters they consider as providing a "quantitative" assessment.**

Response: based on the existent literature, we understand quantitative BBB permeability assessment the one derived from mathematical models able to describe the kinetics of the contrast agents over time and space in mathematical terms. This allows the estimation of the tissue parameters and therefore the quantitative measurement of the BBB permeability (Villringer et al, American Academy of Neurology 2017; Cuenod and Balvay, Diagnostic and interventional Imaging 2013; Gordon et al, Cardiovascular Diagnosis and Therapy 2014). As for the imaging modalities to be used, we are including computed tomography (CT) and magnetic resonance (MRI). We have implemented this information in our manuscript. Page 4; lines 5-15. The measured quantitative parameters will be: permeability surface product (PS) measured in ml.100g.min⁻¹ for CT and the contrast transfer constant (K^{trans}) for MRI measured either in s⁻¹ or min⁻¹.

Regarding the pharmacokinetics models, although most literature yields that the main and most widely used model is the Tofts model (Tofts et al, 1995), other pharmacokinetics models are known to accurately assess quantitative BBB permeability, such as the Johnson–Wilson tissue homogeneity model, the Patlak model or the adiabatic approximation to tissue homogeneity (Cuenod and Balvay, Diagnostic and interventional Imaging 2013). At this early protocol stage, we think it is not appropriate to establish any of them as reference for the SR-MA. Moreover, as stated in the article and in response to question 4 of reviewer 1, we will be addressing relative increases in permeability, which should decrease the impact of heterogeneity in methods. Each of the reported models in the selected papers will be studied and contrasted with the literature to see if it matches our description for quantitative assessment to validate its adequacy for the study.

- 2. In the "Data Management" section, the authors describe how the found studies will be managed. How will the data extracted from eligible studies be handled? Where and how will it be stored? What kind of program(s) will be used to analyze the data?**

Response: we thank the reviewer for this appreciation. The data extracted from the selected articles through the data collection from, will be stored using a specific meta-analyses software, most likely RevMan 5 by the Cochrane library, since to this program is suitable to store, organize and handle MA content. Nevertheless, we are aware of the limitations of this program in some statistical aspects, thus, for the more refined statistical analysis we aim to use other programs, most likely *R* complemented by the meta package. However, we want to clarify that other programs may be used attending to the data obtained in the collection process and the requirements of the analysis, this being the main reason for not including the program to be used in this protocol (we will include this information in the systematic review/meta-analysis manuscript once the statistical analysis are concluded).

- 3. It would be very beneficial if the authors could provide an example of the data collection form with the protocol (see Minor Comments 8-11 below).**

Response: we thank the reviewer for this appreciation. To fulfill the reviewer's request, we include in this re-submission a sample for the data collection form that we aim to use.

- 4. The authors describe how the risk of bias / methodological quality of the eligible studies will be assessed. How will this assessment influence, or be incorporated into, the review? Will studies below a certain quality threshold be excluded from the meta-analysis? Will a sensitivity analysis be performed? Or will the findings of the studies be placed in the context of their methodological quality?**

Response: we thank the reviewer for this question. In a first analysis all studies will be included in the review regardless the risk of bias/methodological quality. Then, we plan to perform a sensitivity analysis excluding those studies with high/critical risk of bias or poor methodological quality. Regarding the reviewer's comment, we have implemented the information in the "Data synthesis" same section. Page 8 lines 38-40.

5. **What criteria will be used to determine whether or not a quantitative meta-analysis will be feasible (page 9, line 55)? Will this be based on statistical heterogeneity, clinical heterogeneity, both, or other factors (see Ioannidis et al., BMJ 2008)?**

Response: we thank the reviewer for this question and the recommended literature along with it. High heterogeneity by itself will not be considered as a factor to not conduct a quantitative meta-analysis since we already expect high heterogeneity due to the different models and imaging techniques that measure BBB permeability. *Ioannidis et al., BMJ 2008* reinforced our idea as they state "Statistical heterogeneity alone is a weak and inconsistently used argument for avoiding quantitative synthesis". Clinical heterogeneity is also expected because although we are going to include only studies on stroke patients, etiology of stroke and the vascular territory involved will vary between studies or even within each study. Thus, we will not consider clinical heterogeneity as a single criterion not to perform the quantitative MA either. Any heterogeneity we find will be recognize and we will try to explain it in any possible way. The main reasons not to conduct a quantitative meta-analysis will be not having enough studies for the propose or not having the appropriate outcome information. We have implemented this information in the manuscript page 9, lines 1-3.

----MINOR----

1. **The systematic review by Ryu et al. (cited on page 5, line 39) dealt with hypoperfusion and penumbral selection for reperfusion therapy in acute stroke, and did not investigate the influence of BBB permeability.**

Response: we thank the reviewer for the appreciation. The reference has been modified accordingly in the manuscript. Page 4; lines 17-18.

2. **How do the authors define "lacunar" and "mild" strokes (page 6, line 28)? Will a size / threshold criterion be used to define a lacunar stroke? Will an NIHSS threshold be used to define mild strokes?**

Response: Lacunar strokes will be defined for patients with a lacunar clinical syndrome and neuro-imaging documenting lacunar infarction (subcortical ischemic lesion with a diameter under 15mm in CT or 20mm in MRI). Mild strokes will be defined mainly according to each article's definition. However, we will define a maximum acceptable cut off of NIHSS below 6. We included this information in Page 5; Table 1.

3. **It's not clear why the authors are planning to exclude "Studies no [sic] reporting contralateral permeability values" if they are only including studies with quantitative BBB permeability values (although this depends on how they define "quantitative", see Major Comment 1).**

Response: we thank the reviewer for this question. We plan to exclude studies not reporting contralateral side because we aim to use the contralateral values as controls for each study. Our aim is to compare the permeability increment (ipsilateral vs. contralateral) within the different studies inside each stage and then compare the total stage increment between stages. If no contralateral values are given, the "control" comparison would not be possible and the heterogeneity in methods would render the results difficult to interpret.

4. **How will studies in languages that the investigators do not understand be dealt with?**

Response: in case a non-understandable language study is obtained, we will consider by its English abstract if the information is interesting enough to be included in the SR. In case it is, the paper will be sent to a professional translator. This information has been implemented on page 5; lines 2-4 of the manuscript.

5. Will the authors include preprints in their search strategies? If so, which databases will be searched for preprints?

Response: no, pre-prints will not be included in the search strategies. This information has been added to the manuscript, page 5; line 17.

6. How will information from unpublished studies found on ClinicalTrials.gov be incorporated into the review?

Response: we thank the reviewer for this important question. If the studies found in ClinicalTrials.gov yield interesting results for the SR-MA, the corresponding author listed in the trial will be contacted to obtain the required information. If no response is given or, if the author decides not to share the data, this will be listed as the reason for exclusion of said trial. We have implemented this information in the manuscript page 5; lines 12-15.

7. The authors might want to consider expanding their search terms to include "leakage", as well as the individual neuroimaging modalities / techniques they are looking for (e.g. "dynamic contrast enhanced MRI", "dynamic susceptibility contrast MRI", "computed tomography perfusion").

Response: we thank the reviewer for this suggestion. Search strategy query has been modified accordingly in the manuscript. Table 2. Pages 5-6.

8. What is meant by "Type of stroke" in Table 3? Does this refer to the stroke etiology? What kind of classification system will be used to assess this (e.g. TOAST)?

Response: Yes, with "type of stroke" we meant stroke etiology and the classification system will be, indeed, the TOAST classification. We thank the reviewer for this comment and in light of it, we modified the manuscript accordingly. Page 7; Table 3.

9. Which "imaging characteristics" (Table 3) will be included?

Response: we thank the reviewer for this comment. For imaging characteristics, we are referring to imaging acquisition parameters and they will vary according to the imaging technique:

- **For the CT modality:** scanner type, length of time of the image series, number and frequency of images, number and thickness of slices, contrast agent, injection rate of contrast agent, tube voltage, tube current.
- **For the MRI modality:** scanner, magnet, MRI protocol, slices, contrast agent, injection rate of contrast agent, flip angle, field of view, matrix, repetition time, echo time.

We detailed these parameters in the data collection form included in this re-submission.

10. By "treatment given" (Table 3), are the authors referring to stroke-specific treatments in the acute phase (i.e. intravenous thrombolysis and mechanical thrombectomy)? Or does this variable also include non-stroke-specific and later treatments (e.g. secondary prevention therapy)?

Response: we thank the reviewer for this question. We are in fact referring to stroke-specific treatments in the acute phase. This information will be collected for subgroup analysis as indicated in the "subgroup" section of this manuscript. This information has been clarified in Page 7; Table 3; "Intervention" section.

11. Which "permeability values" (Table 3) will be extracted (see also Major Comment 1).

Response: as we answered in Major comment 1, the permeability values extracted will correspond to the PS values or K^{trans} values according to what the paper reports. Ipsilateral and contralateral mean

values and their respective standard deviations will be collected. When extracting these values, the pharmacokinetic model used and the characteristics of permeability assessment (software, ROI etc.) will be extracted too.

Details can be found in the data collection form included in this re-submission.

12. Will the grouping based on time from onset to imaging (page 8, lines 53-57) be done by study or within each study as well (if this information is available)?

Response: we thank the reviewer for this question. We will conduct both grouping strategies, as available. As there is a great lack of human studies reporting BBB dynamics through stroke stages, we expect that the vast majority of studies will only report one scan time at which permeability was assessed, then, according to that scan time the study will be placed in the corresponding group. Nonetheless, we are aware of some studies reporting more than one scan time (BBBP assessment through time). For those, the values reported in each time will be considered as separate studies and each time will be placed in the corresponding group.

In light of the reviewer comment, we implemented this information in page 7; lines 27-30 of the manuscript.

13. Will statistical tests be used in addition to the funnel plot to assess publication bias?

Response: we thank the reviewer for this comment. In addition to the funnel plot, the Egger's test to quantify the funnel plot's asymmetry will be performed. We implemented this information accordingly in the reviewed manuscript on the "Publication bias" section. Page 9; lines 15-16.

VERSION 2 – REVIEW

REVIEWER	Richard Leigh Johns Hopkins University, USA
REVIEW RETURNED	02-Jul-2020

GENERAL COMMENTS	The revised version addressed the issues raised. The proposed study will be a challenging endeavor, however if done successfully, could have a significant impact.
--

REVIEWER	Ahmed Khalil Charité Universitätsmedizin Berlin, Germany Berlin Institute of Health, Germany Max Planck Institute for Human Cognitive and Brain Sciences, Leipzig, Germany
REVIEW RETURNED	13-Jun-2020

GENERAL COMMENTS	I would like to thank the authors for the effort they've put into revising this manuscript. They have addressed all my concerns, with one exception: the exact way in which the methodological quality and/or risk of bias of the studies will be used to exclude/include studies should be clearly explained in the study protocol. If the authors will be "excluding those studies with high/critical risk of bias or poor methodological quality", then I assume a score is going to be calculated (e.g. from the checklists / tools mentioned in the manuscript). If so, what will be the cutoff for excluding studies? It is imperative that this be defined by the authors prior to conducting the systematic review, otherwise the risk of selection bias in the review will be large.
--

VERSION 2 – AUTHOR RESPONSE

Reviewer: 1

Reviewer Name: Richard Leigh

Institution and Country: Johns Hopkins University, USA

Please state any competing interests or state 'None declared': None declared

The revised version addressed the issues raised. The proposed study will be a challenging endeavor, however if done successfully, could have a significant impact.

Response: we deeply thank the reviewer for the effort put on improving our manuscript and for the publication recommendation.

Reviewer: 2

Reviewer Name: Ahmed Khalil

Institution and Country: Charité Universitätsmedizin Berlin, Germany; Berlin Institute of Health, Germany; Max Planck Institute for Human Cognitive and Brain Sciences, Leipzig, Germany

Please state any competing interests or state 'None declared': None declared

I would like to thank the authors for the effort they've put into revising this manuscript. They have addressed all my concerns, with one exception: the exact way in which the methodological quality and/or risk of bias of the studies will be used to exclude/include studies should be clearly explained in the study protocol. If the authors will be "excluding those studies with high/critical risk of bias or poor methodological quality", then I assume a score is going to be calculated (e.g. from the checklists / tools mentioned in the manuscript). If so, what will be the cutoff for excluding studies? It is imperative that this be defined by the authors prior to conducting the systematic review, otherwise the risk of selection bias in the review will be large.

Response: we deeply thank the reviewer for the effort put on improving our manuscript and for the publication recommendation.

In regard to the question raised: indeed, a score will be calculated according to the proposed tools.

- **NOS scale:** This scale proposes a 'star system' in which a high-quality choice will be granted by a star. NOS comprise 4 categories with 8 items, and a maximum of 9 stars for high-quality study can be given. Since currently there is no standard criterion for what can constitute a high-quality study using this scale, we base our criteria on literature (**Islam et al. *Neuroepidemiology* 2016; Luchini et al. *World J Meta-Anal* 2017**). Thus, a score of 7 or more corresponds to a high-quality study/low risk of bias and less than 5 will be high or critical risk of bias/poor quality. Therefore, studies below 5 will be excluded in the sensitivity analysis.
- **Cochrane Collaboration's tool for assessing risk of bias in randomized trials:** The risk of bias for each domain composing this scale will be graded high, low or unclear based on the relevant information extracted from each eligible study. According to the guidelines of Cochrane, low risk of bias will be given if the trial has been graded in all of the scale sections as low risk, unclear risk of bias will be given when some concerns appear at least in one of the domains, and high risk of bias will be given to the study if this grade is given to at least one of the domains of the check list. Then, any study with at least 1 section of the checklist graded as high risk, will be excluded in the sensitivity analysis.

This information has been implemented in the manuscript in Page 8, lines 11-15 and 19-22.

VERSION 3 – REVIEW

REVIEWER	Ahmed Khalil Charité Universitätsmedizin Berlin, Germany
REVIEW RETURNED	16-Jul-2020
GENERAL COMMENTS	No further comments. Thank you.